# Diagnostic Testing in Suspected Primary Mitochondrial Myopathy

Jose C. Hinojosa [1] and Salman Bhai [1,2,*]

[1] Neuromuscular Center, The Institute of Exercise and Environmental Medicine (IEEM), 7232 Greenville Ave, Dallas, TX 75231, USA

[2] Neurology Department, University of Texas Southwestern Medical Center (UTSW), 5323 Harry Hines Blvd, Dallas, TX 75390, USA

* Correspondence: salman.bhai@utsouthwestern.edu

**Abstract:** The diagnosis of primary mitochondrial myopathy is often delayed by years due to non-specific clinical symptoms as well as variable testing of mitochondrial disorders. The aim of this review is to summarize and discuss the collective findings and novel insights regarding the diagnosing, testing, and clinical presentation of primary mitochondrial myopathy (PMM). PMM results from a disruption of the oxidative phosphorylation (OXPHOS) chain in mitochondria due to mutations in mitochondrial DNA (mtDNA) or nuclear DNA (nDNA). Although there are many named syndromes caused by mitochondrial mutations, this review will focus on PMM, which are mitochondrial disorders mainly affecting, but not limited to, the skeletal muscle. Clinical presentation may include muscle weakness, exercise intolerance, myalgia, and rhabdomyolysis. Although skeletal muscle and respiratory function are most frequently affected due to their high energy demand, multisystem dysfunction may also occur, which may lead to the inclusion of mitochondrial myopathies on the differential. Currently, there are no effective disease-modifying treatments, and treatment programs typically only focus on managing the symptomatic manifestations of the disease. Although the field has a large unmet need regarding treatment options, diagnostic pathways are better understood and can help shorten the diagnostic journey to aid in disease management and clinical trial enrollment.

**Keywords:** mitochondrial myopathy; genetics; myopathy; exercise intolerance; fatigue

## 1. Introduction

Metabolic and mitochondrial myopathies are a rare set of genetic disorders that involve the impairment of energy metabolism. Such disruptions can be found during carbohydrate metabolism (glycogen-storage disease), fatty acid oxidation (fatty acid oxidation disorder), or the mitochondrial respiratory chain (mitochondrial myopathy) [1]. The aim of this review is to focus on the latter, where the primary defect is due to the dysfunction of the electron transport chain (ETC). Primary mitochondrial myopathy (PMM) results from defects in nuclear or mitochondrial DNA, leading to direct or indirect dysfunction of the respiratory chain. This causes decreased aerobic energy production, amongst several other possible mechanisms, and thus, muscle disease [1]. Although multisystem manifestations may occur from tissues with high metabolic demand, the term primary mitochondrial myopathy is used when skeletal muscle is the predominantly affected tissue [2].

The mitochondria are double-membraned cellular organelles that are responsible for producing over 90% of the adenosine triphosphate (ATP) necessary to maintain the proper functioning of tissues [3]. Additionally, mitochondria are key regulators for programmed cell death (apoptosis), $Ca^{2+}$ homeostasis, biosynthesis of heme, and β-oxidation of fatty acids, and are the site for numerous other metabolic processes [4,5].

Embedded within the inner membrane (IM) are five multicomplex proteins (Complex I–V), collectively part of the electron transport chain [4]. As electrons are transferred from

complex to complex, protons are accumulated in the inner membrane space, resulting in the electrochemical charge across the IM that drives ATP production. However, during this process, specifically known as oxidative phosphorylation (OXPHOS), many unpaired electrons escape the ETC as free radicals. This free radical production is a normal part of healthy respiration but is produced in excess in dysfunctional mitochondria. This excess production of free radicals can further contribute to mitochondrial dysfunction, resulting in disease [4–6].

## 2. Energetics

To better understand the presentation of PMM, knowledge of metabolic pathways leading up to mitochondrial dysfunction, specifically during the rest-to-active transitions, can help contextualize clinical presentations of metabolic and mitochondrial myopathies.

At rest, the primary fuel sources are fatty acids (FA); however, the transition from rest-to-active requires immediate utilization and additional production of ATP [1]. The intensity and duration of this transition, and the activity following, will dictate whether ATP is produced anaerobically or aerobically [1]. Although a complete review of these two processes is beyond the scope of this paper, a brief summary will be provided.

As a framework for these metabolic processes, they will be discussed as if they occur separately and linearly; however, there is significant overlap between the ATP production pathways and not one single process is solely responsible for ATP production at a given time.

### 2.1. Phosphocreatine System

For the short-duration activity lasting 8–10 s, the anaerobic production of ATP occurs through the phosphocreatine system (ATP-PC system), which is the fastest pathway to generate ATP [1,7]. This pathway is limited by the phosphocreatine (PC) stores, and, once depleted, the other metabolic pathways are activated to produce ATP. This process is not a typical site of primary pathology.

### 2.2. Glycolysis

The second pathway is anaerobic glycolysis, which involves the metabolism of glucose to produce ATP. As PC stores deplete, glycolysis begins to supplement ATP production. Due to its slower kinetics, the onset of glycolysis begins after about 10 s of exercise and can last up to two and a half minutes [1,7]. The other primary product of glycolysis is lactic acid, which can accumulate and lead to a decrease in pH and muscle contractility. Disruptions in glycolysis and the upstream production of glucose from glycogen breakdown fall in the category of glycogen storage diseases (GSDs) and, more broadly, glycogenoses.

### 2.3. Oxidative Phosphorylation

The final and most efficient metabolic pathway is OXPHOS. This pathway utilizes oxygen to produce ATP. After glycolysis, pyruvic acid is then converted into acetyl-CoA, which enters the mitochondria.

While in the mitochondrion, acetyl-CoA reacts with oxaloacetate to produce nicotinamide adenine dinucleotide (NADH) and flavin adenine dinucleotide (FADH$_2$) biproducts (throughout the citric acid cycle) that donate their electrons and hydrogens for the electron transport chain (ETC) [1]. The electrons and hydrogens are passed down the ETC from carrier to carrier and, finally, to oxygen, which is the final hydrogen acceptor. It is here that the bulk of the ATP is produced through OXPHOS [1,7].

With much slower kinetics than the anaerobic pathways, OXPHOS does not begin until after about two and a half minutes and can last up to two and a half hours in the moderate intensity domain. OXPHOS can also utilize FA to produce acetyl-CoA through β-oxidation, and amino acids to produce intermediates anywhere along the OXPHOS metabolic process. The maximal level of oxygen utilization defines the aerobic capacity of an individual (VO$_2$ max).

Metabolic and mitochondrial myopathies can occur from dysfunction in any component of the pathways discussed above [1]. Depending on where this dysfunction occurs, the myopathy is categorized accordingly. Disorders of glycogen or glucose degradation (glycogenolysis or glycolysis, respectively) or glycogen synthesis (glycogenesis) lead to abnormal accumulation of glycogen and are termed GSDs. These typically present as repeated episodes of rhabdomyolysis and cramping, although the spectrum of presentation depends on the exact type of GSD.

Given the reliance on the glycolytic system early in activity, patients have exercise intolerance during the initial phase of exercise, especially to high-intensity exercises. This contrasts with disorders of lipid metabolism and mitochondrial myopathies, which present with exercise intolerance to low-intensity, endurance activities, and rhabdomyolysis. Mitochondrial dysfunction can result from various defects along the OXPHOS pathway, including those that affect respiratory chain proteins, which will be the following focus of this review [1,7].

### 3. Mitochondrial Genetics

Every cell, except red blood cells, contains mitochondria, and each mitochondrion contains numerous genomes. Most of the genes required for proper mitochondrial maintenance are dispersed throughout nuclear DNA (nDNA), with nearly 1700 mitochondrial proteins that are encoded by nDNA [8]. However, the mitochondrion itself has its own genetic material, known as mitochondrial DNA (mtDNA) [6]. In contrast to nDNA, mtDNA is small (16,569 base pairs), circular, and located in the mitochondria [8]. The mtDNA contains 37 genes, in which 24 of those genes are responsible for the synthesis of the protein subunits in the ETC complexes [2,8]. Mitochondrial disorders stem from mutations in genes that encode for ETC complexes, as well as from mutations in genes necessary for their translation and assembly [3].

While nDNA follows Mendelian inheritance patterns, mtDNA is maternally inherited [9]. From this pattern of inheritance, any mutated mtDNA in the mother will be passed on to her offspring, while mutated mtDNA in the father will not [2]. Although this is a widely agreed-upon concept, recent studies contend that mtDNA may also be paternally inherited; however, this requires further investigation [10,11].

Although healthy individuals may contain a low percentage of mutant mtDNA (typically <1%), clinical relevance occurs when the mitochondria contain a greater proportion of mutated DNA than wild-type (normal) DNA [12]. This phenomenon, known as heteroplasmy, exists where a proportion of genomes contain mutations, and a proportion are wild-type. Such abnormalities can include point mutations, single large-scale deletions, depletion, and multiple deletions [6]; however, this list is not exhaustive. Other factors that can contribute to a greater percentage of mutant mtDNA include, but aren't limited to, aging, illness, and inflammation [6].

mtDNA may also exist in a state of homoplasmy, meaning all copies are identical within a mitochondrion [13]. Homoplasmic mtDNA point mutations often cause mild defects that affect one organ or tissue (i.e., deafness or cardiomyopathy) [13] but can also be associated with more severe, multisystem, disease presentation (i.e., Leber's hereditary optic neuropathy). In contrast, heteroplasmic mutations more often involve multisystem dysfunction, particularly in the skeletal muscle, spinal cord, heart, brain, and endocrine organs due to their large energy demand [6,7].

The degree of heteroplasmy often dictates the severity of PMM [12]. A higher percentage of mutant heteroplasmy correlates with a younger age of onset and a more severe disease phenotype [4,5]. Several sources describe a threshold of 80–90% mutant mtDNA in mitochondria in which disease manifestation occurs [9], while others cite a lower percentage of around 60–80% [13]; however, overt mitochondrial disease has been seen in far lower percentages of mutant mtDNA, indicating an unclear threshold for disease manifestation.

Although extreme percentages (very high or very low) often correlate with disease severity, intermediate levels of heteroplasmy produce higher variability in phenotypic

presentation of PMM [12], where lower levels of mutation load cause one type of clinical presentation and higher levels cause another [3].

Furthermore, due to the complex nature of transmitting mtDNA mutations, it is not currently possible to predict the level of heteroplasmy that women with mtDNA mutations will pass on to their offspring. The poor genotype-phenotype correlation, where the same mutation in family members can have a range of clinical phenotypes, complicates matters. Although the phenotype of PMM is unpredictable and variable, there are some key clinical features to consider when it comes to diagnosing PMM.

## 4. Clinical Presentation

PMM is a progressive disease that is both genetically and phenotypically variable and can manifest in a variety of ways. Although originally thought to be extremely rare, mtDNA disorders currently affect approximately 1 in 4300 of the population [14]. Today, mutations in over 350 genes in both mitochondrial and nuclear genomes affecting multiple aspects of mitochondrial dynamics have been identified as causing PMM [2,6], along with a wide range of overlapping clinical phenotypes [12]. Table 1 below offers some examples of genes causing mitochondrial disorders.

**Table 1.** Features of select mitochondrial disorders.

| Disease | Genes | Mutation Mechanism | Clinical Features | Inheritance |
|---------|-------|--------------------|-------------------|-------------|
| CPEO [15] | *POLG1, 2, ANT1, OPA1, MPV17, TK2* | mtDNA replication maintenance and transcription | Ptosis, myopathy, bilateral ophthalmoparesis | Autosomal dominant, recessive, sporadic, or maternal |
| MELAS [16] | *MT-TL1, MT-ND1, 5* | mt tRNA and OXPHOS CI subunit | Stroke-like episodes, myopathy, encephalopathy, seizures | Maternal |
| KSS/PS | Can be anywhere in the mitochondrial genome | mtDNA single or large-scale deletions (~2–10 kb of mtDNA) | CPEO, myopathy, onset before 20 years, cardiomyopathy | Maternal, sporadic |
| MERRF [17] | *MT-TK* | mt tRNA | Generalized epilepsy, ataxia, myopathy, ragged red fibers | Maternal, sporadic |
| CoQ10 deficiency [18] | *COQ2, 4, 6, 7, 8A, 8B, 9, PDSS1, 2* | OXPHOS electron transporter | Early infantile onset encephalopathy | Autosomal recessive |
| TK2 deficiency [19] | *TK2* | Mitochondrial maintenance defect; nucleotide pool maintenance | Infantile onset myopathy, proximal weakness | Autosomal recessive |
| LHON [20] | *MT-ND1, 4, 4L, 6* | OXPHOS CI subunit mutation | Optic atrophy, possible multisystem involvement, cardiomyopathy | Maternal |
| LS [21] | *SURF1, MRPS34, MT-ND3, MT-ATP6* | OXPHOS CI and CV subunits, mt ribosomes, and CIV assembly | Developmental delay, regression, ataxia, respiratory abnormalities | Autosomal recessive |

Examples of genes leading to mitochondrial disorders (adapted from [3,12]). This list is not exhaustive. These disorders result in various degrees of OXPHOS dysfunction depending on the mutation. For a more comprehensive list please visit MITOMAP. CPEO: chronic progressive external ophthalmoplegia (OMIM ID: #157640), MELAS: mitochondrial encephalomyopathy, lactic acidosis, and stroke-like episodes (OMIM ID: #540000), KSS/PS: Kearns-Sayre syndrome/Pearson syndrome (OMIM ID: #530000), MERRF: myoclonus epilepsy and ragged red fibers (OMIM ID: #545000), LS: Leigh Syndrome (OMIM ID #256000), LHON: Leber's hereditary optic neuropathy (OMIM ID: #308905), TK2: Thymidine Kinase 2 (OMIM ID *188250), CoQ10: Coenzyme Q10 (OMIM ID: #607426).

PMM is a highly variable disorder that exists on a spectrum, ranging from adult-onset single-organ system involvement to infantile-onset multisystem dysfunction and lethal disease [22]. Although named or multisystem disorders (i.e., mitochondrial encephalomyopathy with lactic acidosis and stroke-like episodes (MELAS), myoclonic epilepsy with ragged red fibers (MERRF), or Kearns-Sayre/Pearson syndrome: OMIM ID #530000, etc.) [1,5,23] may include myopathy, this clinical feature is often overshadowed by other symptoms.

Although a range of symptoms within a clinical presentation may exist in PMM, there are prominent features to consider when evaluating patients for mitochondrial myopathy. Additionally, PMM is a progressive disease, with symptoms worsening over time, and with the possibility of newer symptoms developing as the disease progresses [24].

### 4.1. Exercise Intolerance

Most individuals with PMM will have limited exercise capacity due to a low $VO_2$max (maximal amount of oxygen consumption during exercise) [8,22]. However, this decreased ability to utilize oxygen extends beyond their exercise performance and into their activities of daily living, such as walking, cooking, cleaning, and grocery shopping [22]. This typically presents as premature physical fatigability, which is often disproportional to the degree of muscle weakness [23].

This physical fatigue can be defined as the inability to sustain muscle contraction or exertion during aerobic exercise [22], which varies from patient to patient and depends on mutation load. On a single-limb scale, fatigue may be induced by restricted ATP production, depletion of PC and glycogen stores, as well as increased lactate production; on a whole-body scale, fatigue may be due to factors such as impaired oxygen uptake, increased carbon dioxide levels, and increased lactate production [22].

Additionally, patients with severe levels of mtDNA mutations will display exercise intolerance due to the muscle's inability to increase oxygen utilization in relation to physiological responses that increase oxygen delivery. Also, patients with PMM had significantly lower peak-work capacities and oxygen uptake when compared to healthy controls [22].

They also discovered an inverse relationship between the mutation load in skeletal muscle and peak extraction of oxygen during exercise, indicating that the degree of exercise intolerance correlates largely with the severity of impaired oxidative phosphorylation due to the muscle's impaired ability to extract and utilize oxygen [25].

In healthy individuals, the decreased ability to produce additional force at maximal exercise capacity is recognized as a safety mechanism to avoid tissue damage [22]; however, in patients with PMM, this subjective sense of fatigue occurs before most patients begin to approach maximal capacity. While data on exercise recovery is unclear, patients with PMM appear to have prolonged recovery periods in the clinical setting, though evidence is needed to support this. Other symptoms of exercise intolerance include elevated baseline cardiac output ($Q_c$) and dyspnea (shortness of breath) [1]. Such symptoms may even be exacerbated during periods of stress, such as injury, infection, fasting, or long-duration exercise [1,8].

### 4.2. Proximal and/or Axial Weakness

Proximal weakness is a common feature in patients with mitochondrial disorder. The degree of weakness is variable and can extend to other muscles in the face, neck, arms, and/or legs. Additionally, in some cases, muscles of the diaphragm or respiratory system can be affected, potentially requiring the need for ventilator support [9,26].

### 4.3. Exercise-Induced Muscle Pain and Rhabdomyolysis

In conjunction with exercise intolerance and muscle weakness, exercise-induced muscle pain (myalgias) is another common feature reported by those affected with PMM [25]. This is often reported as burning or leg heaviness during long-duration exercise or even with simple activities. True muscle cramps are not often developed in individuals with PMM [1]. Rhabdomyolysis can occur, although it is not common, and can be triggered by exercise or illness.

### 4.4. Ocular Abnormalities

Weakness often begins with the muscles of the eyes and eyelids, leading to chronic progressive external ophthalmoplegia (CPEO). Ophthalmoplegia, and/or ptosis, can be one of the first symptoms of PMM [9] and is characterized by symmetric, progressive, bilateral

paresis of extrinsic ophthalmic muscle. There is often a compensatory contraction of the frontalis muscle, with a head tilt and the need to turn to see [27]. Patients initially may complain of blurry vision, and close examination will reveal limited extraocular movements.

CPEO can occur as an isolated symptom or part of multisystem dysfunction, such as in Kearns–Sayre syndrome (KSS) and Pearson syndrome (PS), which are part of a spectrum of disorders with CPEO being the mildest, and PS being the most severe [27]. These disorders typically are a result of a large-scale mtDNA deletion, which can be related to nDNA mutations (e.g., *TK2* and *POLG*) [2]. In CPEO, proximal myopathy is not necessary, and patients may simply have extraocular movement abnormalities or ptosis.

Although these are common features of PMM, it is important to recognize that fatigue, exercise intolerance, generalized weakness, and muscle pain are all nonspecific symptoms that can be a result of many other disorders other than PMM, such as cardiopulmonary disease, medication use, deconditioning, or psychological disorders. Thus, a careful and thorough workup of the patient history is necessary to increase suspicion of PMM and evaluate for other disorders that may explain the symptoms.

## 5. Diagnostic Approach

Diagnosing mitochondrial myopathies should involve a systematic approach, ultimately seeking a molecular etiology [6,8]. Every work-up should be carefully tailored to each individual presenting with PMM features to ensure an accurate diagnosis. Figure 1 below shows the authors' approach, which can be used to efficiently evaluate a patient with a suspected mitochondrial myopathy.

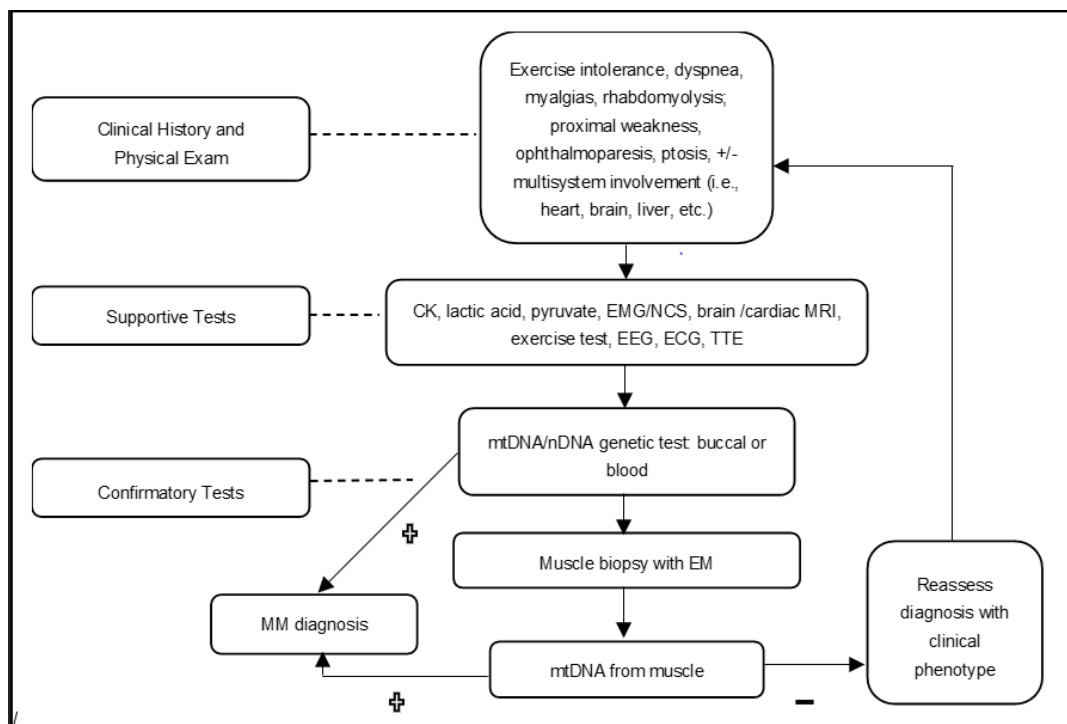

**Figure 1.** Diagnostic template that may be used when treating suspected PMM. CK: creatine kinase, EMG/NCS: electromyogram and nerve conduction study, MRI: magnetic resonance imaging, EEG: electroencephalogram, ECG: electrocardiogram, TTE: transthoracic echocardiogram, and EM: electron microscopy.

Diagnostic testing can fall into two broad categories. The first of which identifies phenotypic presentation, and excludes other disorders, to work towards a PMM diagnosis. These types of diagnostic tests confirm the presence of dysfunction in affected organ systems, rather than definitively confirming a diagnosis of mitochondrial myopathy, and are often supplemental to the clinical history and physical examination [6]. Such tests

involve, but are not limited to, cardiac and endocrine investigations, EMG, blood serum studies (i.e., creatine kinase (CK), lactic acid, or amino acids), or evaluating hearing or visual symptoms [1,6,22].

The second category of tests directly assesses whether a patient is affected by PMM. These mainly consist of genetic testing and muscle biopsy [6]. However, neither category of testing should solely be relied upon, as a thorough combination of both is needed when considering a diagnosis of PMM. Table 2 below includes supportive tests that can be used when making the case for PMM.

**Table 2.** Clinical symptoms and tests for mitochondrial disorders.

| Signs/Symptoms | Tests | Possible Results |
|---|---|---|
| Seizures, encephalopathy | EEG | Epileptiform activity, diffuse slowing |
| Stroke-like episodes, ataxia, developmental delay, dementia, seizures | Brain MRI | Variable findings |
| Neuropathy | NCS/EMG | Axonal sensory or sensorimotor neuropathy |
| Myopathy | NCS/EMG, CK | Normal or slightly elevated. Elevated in $CoQ_{10}$ deficiency |
| Cardiac dysfunction | ECG, TTE, cardiac MRI | Conduction abnormalities, cardiomyopathy |
| Cognitive dysfunction | Neurocognitive testing | Impaired executive functioning, memory, auditory/verbal learning |
| Ocular abnormalities | EOM exam, Ophthalmology referral/exam | Optic atrophy, retinopathy, ptosis, ophthalmoplegia |
| Pulmonary dysfunction | PFTs | Decreased FEV1 and FVC |
| Biochemical markers | Serum and CSF lactate, LFTs, serum amino acids | Normal or elevated lactic acid/LFTs, alanine can be elevated |
| Endocrine dysfunction | HgbA1c, TSH, PTH, cortisol | Reduced cortisol, elevated HgbA1c and PTH, variable TSH |
| Dysphagia | Swallow studies | dysmotility, aspiration |
| Exercise intolerance | Exercise test | Elevated $Q_c/VO_2$ |

Diagnostic tests and corresponding symptoms when assessing for PMM (adapted from [6]). EEG: electroencephalogram, MRI: magnetic resonance imaging, EMG/NCS: electromyogram and nerve conduction study, CK: creatine kinase, ECG: electrocardiogram, TTE: transthoracic echocardiogram, EOM: extraocular movement, PFTs: pulmonary function tests, FEV1: forced expiratory volume in one second, FVC: forced vital capacity, CSF: cerebrospinal fluid, LFTs: liver function tests, HgbA1c: hemoglobin A1c, PTH: parathyroid hormone, and TSH: thyroid stimulating hormone, $Q_c/VO_2$: ratio between cardiac output and oxygen consumption.

*5.1. Laboratory Studies*

Once a mitochondrial myopathy is suspected, clinical testing (Table 1) of dysfunction in organ systems based on clinical presentation is typically undertaken to support the diagnosis. This helps exclude other etiologies of dysfunction and also aids in the management of involved organ systems. Although these tests may be helpful, it is important to note that they are neither sensitive nor specific and can sometimes produce false positives, or even appear normal [1,28]. For example, elevated CK levels may be caused by exercise, statin use, or normal variations, while serum lactic acid elevations may occur due to exercise, prolonged tourniquet time, or even technical errors in receiving the sample [1,8]. Both CK and lactic acid can be normal in PMM. If PMM is limited primarily to muscle, several studies, such as brain MRI and CSF studies, are not necessary.

Testing is dictated by symptoms and known progression of disease (e.g., cardiac, ophthalmologic, and endocrine evaluations in CPEO). Other laboratory studies could include a growth differentiation factor 15 (GDF-15) and a fibroblast growth factor 21 (FGF-21); although this is not commonly used in clinical practice and is non-specific for mitochondrial disorders.

### 5.2. Molecular Genetics Studies

Targeted next-generation sequencing (NGS) panel testing is the diagnostic procedure of choice [26], coupled with clinical assessment and other symptom-guided clinical testing. Early genetic testing is advantageous because other genetic diagnoses with shared clinical phenotypes are often included, costs have become more affordable, and it serves as a non-invasive option [8]. Muscle biopsy may not be needed if a pathogenic mutation is found and it matches the clinical phenotype.

Mutations causing PMM can be found in both mtDNA and nDNA. For PMM, pathogenic mutations have been found in all 37 mtDNA genes. nDNA and mtDNA analysis from buccal or blood samples with NGS detects low levels of heteroplasmy, point mutations, and single, large-scale mtDNA deletions.

Although NGS can help identify pathogenic variants, variants of unknown significance (VUS) are common results [1]. VUS resolution will require further clinical testing and a muscle biopsy along with familial genetic testing. Additionally, if an mtDNA abnormality is strongly suspected, and blood and buccal samples yield inconclusive results, analysis of muscle mtDNA is highly recommended, especially for mtDNA deletions.

### 5.3. Electrophysiology

After laboratory screenings, if PMM is suspected but still unclear, electromyography (EMG) can be a useful tool for identifying mimics of PMM [1], or even myotonic disorders and muscular dystrophies [8]. EMG is often normal in mitochondrial myopathies but may show a nonspecific myopathic pattern without muscle membrane irritability. If needed, this test can also aid in determining which muscles are best suited for biopsy [5].

### 5.4. Exercise Testing

Cycle ergometry testing provides unique perspectives to measuring physiologic and biochemical responses to exercise, especially when genetic testing and/or muscle biopsy are inconclusive [1]. The use of cycle testing can be utilized in both the clinical and research settings for mitochondrial myopathy [26]. Since skeletal muscle can be used to assess mitochondrial function, exercise studies can be utilized to observe the physiological consequences of dysfunctional mitochondria. Typically, a standard Bruce protocol is used with a gradual increase in workload on the cycle ergometer [1]. Common measures that are observed in exercise testing include heart rate, blood pressure, $Q_C$, $VO_2$, and oxygen saturation. Clinicians often use this to highlight the increase in muscle metabolite concentration (i.e., lactate and pyruvate) in venous blood with a slow clearance of accumulated plasma lactate during post-exercise recovery [26].

Coupled with these measurements are the measurements taken during the aerobic forearm exercise test, which is also used as a screening tool for PMM [26]. The combined results typically reflect a reduced $VO_2$max, decreased oxygen extraction, high respiratory exchange ratio, and hyperkinetic circulation (elevated baseline cardiac output) in an individual with mitochondrial myopathy [1]. Patients with PMM display an exaggerated ventilation relative to oxygen utilization (higher $VE/VO_2$), an inability to increase a-v$O_2$ difference above 10 mL/dL (80% sensitivity and 100% specificity for attributing low $O_2$ uptake to impaired mitochondrial function), and a $\Delta Q/\Delta VO_2 \geq 7$ (83% sensitivity, 100% specificity) [25]. However, it is important to note that reduced $VO_2$max can be observed in physically inactive individuals who have a decreased level of mitochondrial enzyme activity [1,8].

### 5.5. Muscle Biopsy

The muscle biopsy can be seen as the foundation for diagnosing myopathies; although genetic studies are replacing the need for biopsies [26,27]. Although genetic and clinical testing may be highly beneficial tools in identifying PMM, it is not uncommon for clinicians to have unclear answers after completing them. Given the high levels of heteroplasmy and the wide range of genetic abnormalities causing mitochondrial myopathies, a large

number of PMM cases cannot be solved using the "genetic first" approach. Therefore, more invasive techniques, such as a muscle biopsy, may be a necessary option when mtDNA variants in blood and buccal samples alone yield ambiguous or false-negative results.

Skeletal muscle is primarily affected in individuals with PMM and is therefore the most frequently biopsied tissue, often from the quadriceps muscle [6]. Additionally, mtDNA deletions are more readily detected in muscle than in blood, buccal, and fibroblasts [3,27]. The authors prefer Bergström needle muscle biopsies over surgical biopsies given the high yield of conducting histochemical and molecular analysis on myofibers.

Individuals with PMM typically show alterations in their muscle chemistry that indicate mitochondrial dysfunction (i.e., ragged-red fibers, ragged-blue fibers, COX-negative fibers, or increased lipid droplets), although in some individuals the biopsy may appear normal [6,28]. Electron microscopy is useful to evaluate the ultrastructure of mitochondria, although findings are non-specific. Various abnormalities are found including subsarcolemmal mitochondrial accumulations and abnormalities in size, cristae, or inclusions. Respiratory enzyme complex analysis can be undertaken, although such testing is not widely available, technically difficult, and not required to diagnose patients.

## 6. Pitfalls

Diagnosing and testing for mitochondrial myopathy is complex and controversial, especially when considering which diagnostic approach is most effective and efficient, while still being mindful and considerate of the time and financial burden of patients. Regardless of which diagnostic route is followed, the various tests described above have limitations that may result in findings that incorrectly dissuade clinicians from pursuing a diagnosis of PMM.

While CK and lactic acid are common screening labs, they can be falsely negative or positive due to medications, activity, technical errors, or diseases other than PMM. In addition, given the wide range of genetic abnormalities and heteroplasmy, buccal swabs and blood samples may not adequately test for mtDNA deletions in PMM. This may lead to the false conclusion that normal "mitochondrial genetic studies" from buccal or blood samples rule out mitochondrial disease. Further work-up should include a muscle biopsy if suspicion remains for PMM; however, histopathology and EM can be normal.

While exercise testing can be a useful tool for supporting a diagnosis of PMM, certain measures, such as $VO_2max$, are susceptible to confounding variables. Although $VO_2max$ is an important indicator of oxidative capacity, this measure is affected by other factors such as deconditioning, effort, and comorbid disease (i.e., diabetes or cardiopulmonary disease) [22], leaving exercise testing inconclusive. Lastly, false positives are possibilities to consider. Multiple mtDNA deletions can be due to age-related changes. While each test individually is not without its limitations and should not be solely relied upon, a careful investigation should be sufficient to accurately diagnose PMM [8,12,22,28].

## 7. Conclusions

Mitochondrial myopathies are a subset of highly heterogenous, progressive diseases caused by the dysfunction of OXPHOS, leading to exercise intolerance, proximal myopathy, pre-mature fatigue, CPEO with or without ptosis, and/or myalgias. Diagnosis is typically based on a variable combination of clinical presentation, laboratory studies, EMG, muscle biopsy, and genetic workup. Genetic testing is supplanting the use of muscle biopsy, although biopsies are still useful in resolving VUS and finding mtDNA mutations due to heteroplasmy. Currently, there are no disease-modifying treatments available to halt the progression of mitochondrial myopathies. Rapid diagnosis can help patients understand their symptoms, enroll in trials, improve disease management, and reduce costs from unnecessary tests.

**Author Contributions:** Conceptualization and Writing: J.C.H. and S.B.; research: J.C.H. and S.B. All authors have read and agreed to the published version of the manuscript.

**Funding:** This research received no external funding.

**Institutional Review Board Statement:** Not applicable.

**Informed Consent Statement:** Not applicable.

**Data Availability Statement:** Not applicable.

**Conflicts of Interest:** The authors declare no conflict of interest.

**Abbreviations**

ATP: adenosine triphosphate, Ca: calcium, CK: creatine kinase, CPEO: chronic progressive external ophthalmoplegia, ECG: electrocardiogram, EEG: electroencephalogram, EM: electromyography, EMG: electromyography, ETC: electron transport chain, FA: fatty acid, GSD: glycogen storage disease, IM: inner membrane, LHON: Leber's hereditary optic neuropathy, MELAS: mitochondrial encephalopathy, lactic acidosis, and stroke-like episodes, MERRF: myoclonic epilepsy and ragged red fibers, MRI: magnetic resonance imaging, mtDNA: mitochondrial DNA, NCS: nerve conduction study, nDNA: nuclear DNA, NGS: next-generation sequencing, OXPHOS: oxidative phosphorylation, PC: phosphocreatine, PMM: primary mitochondrial myopathy, $Q_c$: cardiac output, TTE: transthoracic echocardiogram, VUS: variance of unknown significance

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
