# Peer review of "Diagnostic Testing in Suspected Primary Mitochondrial Myopathy"

_muscles, doi:10.3390/muscles2010007_

Round 1
Reviewer 1 Report
The authors are proposing a review focused on primary mitochondrial myopathies. They summarized the current knowledge in term of diagnosis and clinical testing. They complied the various phenotypes observed in patients and outlined that the tests that could be conducted.
Overall, this review is very well written. The text is clear and the organization makes this review easy to read. The reminder regarding mitochondrial metabolism at the beginning makes this review accessible for non-expert readers.
Other reviews on that sort tend to be out of date and this review give a nice update of the field.
I only have minors comments and couple of suggestions that the authors might choose to follow or ignore:
- PMM is defined in the Abstract, not the main text (line 54). - defined only later line 157
- The main text uses at the beginning MM – mitochondrial myopathy – while the abstract uses PMM. But later in the manuscript, the authors uses PMM. I would suggest to pick one (MM or PMM) and stick to it.
- Define nDNA (line 114) – only defined in Abstract
- Line 122, the authors state “mtDNA is maternally inherited”. I would suggest to mention and explain the recent controversial reports made by Luo S, et al. (2018) and Wei W. Et al. (2020 - 10.1038/s41467-020-15336-3), and further dicussed by several papers such as Pagnamenta et al. (2021 - https://doi.org/10.1038/s41576-021-00380-6) regarding paternally inherited mitochondrial DNA.
- Lines 124 and 125, the authors state “Such abnormalities can include point mutations, single large-scale deletions, depletion, and multiple deletions [5].”. Other types of variants have also been reported. See an exmaple on inversion here: 10.1086/302927.
- I am not sure I agree with the statement “Homoplasmic mtDNA point mutations typically cause mild defects” (line 134). Can the authors elaborate on that?
- I would suggest to add the OMIM ID for each syndrome cited in the paper.
- Lines 160-161: which types of mutations?
- Line 205: TK2 and POLG – “and” should not be italicized
- Line 205: “In CPEA, proximal myopathy is not necessary, 205
and patients may simply have extraocular movement abnormalities or ptosis.” - I assume the authors meant CPEO?=> CEPO? - In the PDF I have got, the Figure 1 is truncated “mtDNA/nDNA genetic test: buccal or”??? and “mtDNA from muscle” at the bottom is barely readible.
- Line 244: “This aids in excluding other etiologies of dysfunction as well as helps with management of involved organ systems.:” - I assume the authors meant “This aids is”?
- Line 228 – define CK
- The authors are using a lot of abbreviations. A list of them could help the reader.
- The first paragraph of the intruduction could use more reference such as “Although multisystem manifestations may occur from tissues with high metabolic demand, the term primary mitochondrial myopathy is used when skeletal muscle is the predominantly affected tissue.” (lines 36-38)
- OXPHOS is re-defined line 83, but it was already defined line 48
- The authors could have present a little more in details or give some examples of genes causing in PMM.
Reviewer 2 Report
The review is dedicated to primary mitochondrial myopathy (PMM) diagnostic testing. The authors briefly introduced the key aspects of mitochondria-related cellular processes and mitochondrial genetics. They also summarized the information about the clinical presentation of PMM and methods of diagnostic testing. However, there are some questions that are not discussed in the review. One of this questions is dedicated to the target of mtDNA or nDNA mutations – are there mutations that cause myopathy by affecting not OXPHOS itself, but mitochondrial biogenesis/mitophagy and fusion/fisson? Are their clinical presentations different from mutations affecting OXPHOS genes? It also seems to me that the section about exercise intolerance could be broadened, that would significantly straighten the review. For example, questions that can be answered are – which type of the exercises cause the most severe muscle fatigue (resistance, endurance, weight lifting)? Are there problems with exercise performance or with muscle recovery, or both? Some examples would be great.
Line 86 - flavin adenine dinucleotide (NADH2) – Which is meant here, flavin adenin dinucleotide (FAD) or nicotine adenine dinucleotide (NAD)?
Line 94 – proteins are utilized by OXPHOS? – it is more correct to say that aminoacids, but not proteins, are utilized by OXPHOS
Figure 1 – check for layout of the figure, the text in the lower box is unreadable
Round 2
Reviewer 2 Report
The article is significantly improved and can be accepted in present form
Author Response
Thank you for the helpful comments and helping us improve the paper.